# The impact of pandemic-related worry on cognitive functioning and risk-taking

**Kevin da Silva Castanheira**[1]*, **Madeleine Sharp**[2], **A. Ross Otto**[1]

**1** Department of Psychology, McGill University, Montreal, Canada, **2** Department of Neurology and Neurosurgery, McGill University, Montreal, Canada

* Kevin.daSilvaCastanheira@mail.mcgill.ca

**Data Availability Statement:** The data and analysis scripts are available on the Open Science Framework, accessible with the following link https://osf.io/87wz5/.

## Abstract

Here, we sought to quantify the effects of experienced fear and worry, engendered by the COVID-19 pandemic, on both cognitive abilities—speed of information processing, task-set shifting, and proactive control—as well as economic risk-taking. Leveraging a repeated-measures cross-sectional design, we examined the performance of 1517 participants, collected during the early phase of the pandemic in the US (April–June 2020), finding that self-reported pandemic-related worry predicted deficits in information processing speed and maintenance of goal-related contextual information. In a classic economic risk-taking task, we observed that worried individuals' choices were more sensitive to the described outcome probabilities of risky actions. Overall, these results elucidate the cognitive consequences of a large-scale, unpredictable, and uncontrollable stressor, which may in turn play an important role in individuals' understanding of, and adherence to safety directives both in the current crisis and future public health emergencies.

## Introduction

The COVID-19 pandemic represents a significant threat to the physical, mental, and economic well-being of people globally. While a spate of recent work has documented the direct effects of the pandemic on mental health [1–4], less is known about the consequences it might hold for cognitive functioning. Identifying and understanding these cognitive and behavioural consequences is especially critical as governments continue to face the challenge of controlling the spread of the virus and mitigating its social, economic, and psychological consequences. More than ever, people are being asked to attend to a continuous stream of public health messages, to adhere to government directives and to control their impulses for the sake of the collective good [5]. Seemingly simple decisions—like refraining from sharing a coffee with friends or registering for the vaccine—are the key elements of the global fight against COVID-19, yet concerning trends suggest declining adherence to these regulations and far-from-universal vaccine acceptance [6, 7]. Successful navigation of these situations is thought to rely on a key set of related cognitive processes often referred to as executive functions [8, 9]—broadly defined as monitoring and selection of behaviours in accordance with internal goals [10]. Here, we elucidate the impact of the early stages of the COVID-19 pandemic on both executive function and risky decision-making.

**Funding:** This work was supported by G. W. Stairs Fund, Natural Sciences and Engineering Research Council of Canada Discovery Grant [RGPIN-2017-03918] https://www.nserc-crsng.gc.ca/nserc-crsng/, Social Sciences and Humanities Research Council of Canada grant [430-2020-00518] https://www.sshrc-crsh.gc.ca/home-accueil-eng.aspx and Canadian, Foundation for Innovation Grant [36557] https://www.innovation.ca/ awarded to ARO, as well as Fonds de Recherche du Québec - Santé https://frq.gouv.qc.ca/en/health/ grant awarded to MS. The funders had no role in study design, data collection and analysis, decision to publish, or preparation of the manuscript.

**Competing interests:** The authors have declared that no competing interests exist.

At the same time, a body of work suggests that executive functioning is impaired under conditions of fear and anxiety [11–15] like those reported during the pandemic [16, 17], possibly owing to the processing resources (e.g., working memory) displaced by excessive worry [18]. For instance, anxiety is demonstrated to impair goal-directed cognitive processing thereby increasing the influence of more reflexive responses [19]. However, past work has also shown that some cognitive processes are preserved—or even facilitated—by anxiety [20]. Given the variability in the reported effects of emotional distress, pinpointing the locus of cognitive impairments brought about by pandemic-related worry may be particularly important.

Here, we provide an initial examination of the effects of pandemic-related worry upon cognitive functioning in a large, representative US-based sample, by measuring pandemic-related worry and assessing three distinct facets of cognitive functioning: 1) processing speed, measured with the digit-symbol coding task [21], 2) the ability to shift between multiple task sets, measured using a task-switching paradigm [22] and 3) proactive cognitive control, the ability to utilize contextual, task-related information in accordance with internally maintained goals, measured by the Dot Pattern Expectancy task (DPX) [23]. We chose these three specific cognitive tasks because they index related yet disparate facets of cognitive ability [24], all rely on anxiety-sensitive executive functions (e.g., working memory) [21, 25, 26], were previously validated [27], and administered online allowing for comparisons with pre-pandemic samples.

We administered this online task battery during the early phase of the COVID-19 pandemic in North America, across 3 waves, between April and June of 2020. We measured self-reported pandemic-related worry using the validated "Fear of Coronavirus" questionnaire [28], and subsequently probed the relationship between pandemic worry and cognitive performance across these tasks, controlling for both financial and overall perceived stress levels. We also explored 1) the impact of the pandemic on cognition by comparing performance of participants recruited during the pandemic to that of participants who had been recruited to participate in separate studies before the pandemic but who completed the same online tasks, and 2) the impact of pandemic progression by directly comparing task performance between waves.

Finally, given the importance of risk assessments in a pandemic [29], we examined the possibility that the effects of pandemic-related worry might also extend to individuals' risk-taking. Previous work has found that anxiety engenders an increased perceived likelihood of negative outcomes [30–32], which can manifest as heightened risk aversion [20, 33, 34]. To evaluate this, we probed the effect of pandemic-related worry on decision-making by measuring individuals' risk attitudes in a traditional economic choice task [35, 36]—specifically, whether pandemic-related worry is associated with an overall tendency towards risk aversion, or if this association was selective to choices pertaining to gains or losses [37–39].

## Materials and methods

### Participants

We recruited three samples of 509, 501 and 507 adult participants residing in the US for waves 1 (April 2nd, 3rd & 6th) 2 (April 17th, 20th -23rd) and 3 (June 19th, 22nd) respectively (1517 total), via Amazon Mechanical Turk (MTurk; see Fig 1) [40]. Given the unprecedented nature of the relationship being examined, we planned the samples in advance anticipating data exclusions and ensuring sufficient statistical power (i.e., 90%) to detect modest effect sizes (r > = 0.20) for the relationships between fear of coronavirus (FCQ) scores and the cognitive measures of interest [41]. Participants provided informed written consent and were paid $5 USD for completing the task battery. Our experimental protocol was approved by the McGill University Research Ethics Board-2 and carried out in accordance with institutional guidelines and regulations. We employed strict exclusion criteria on a task-by-task basis to ensure the quality of

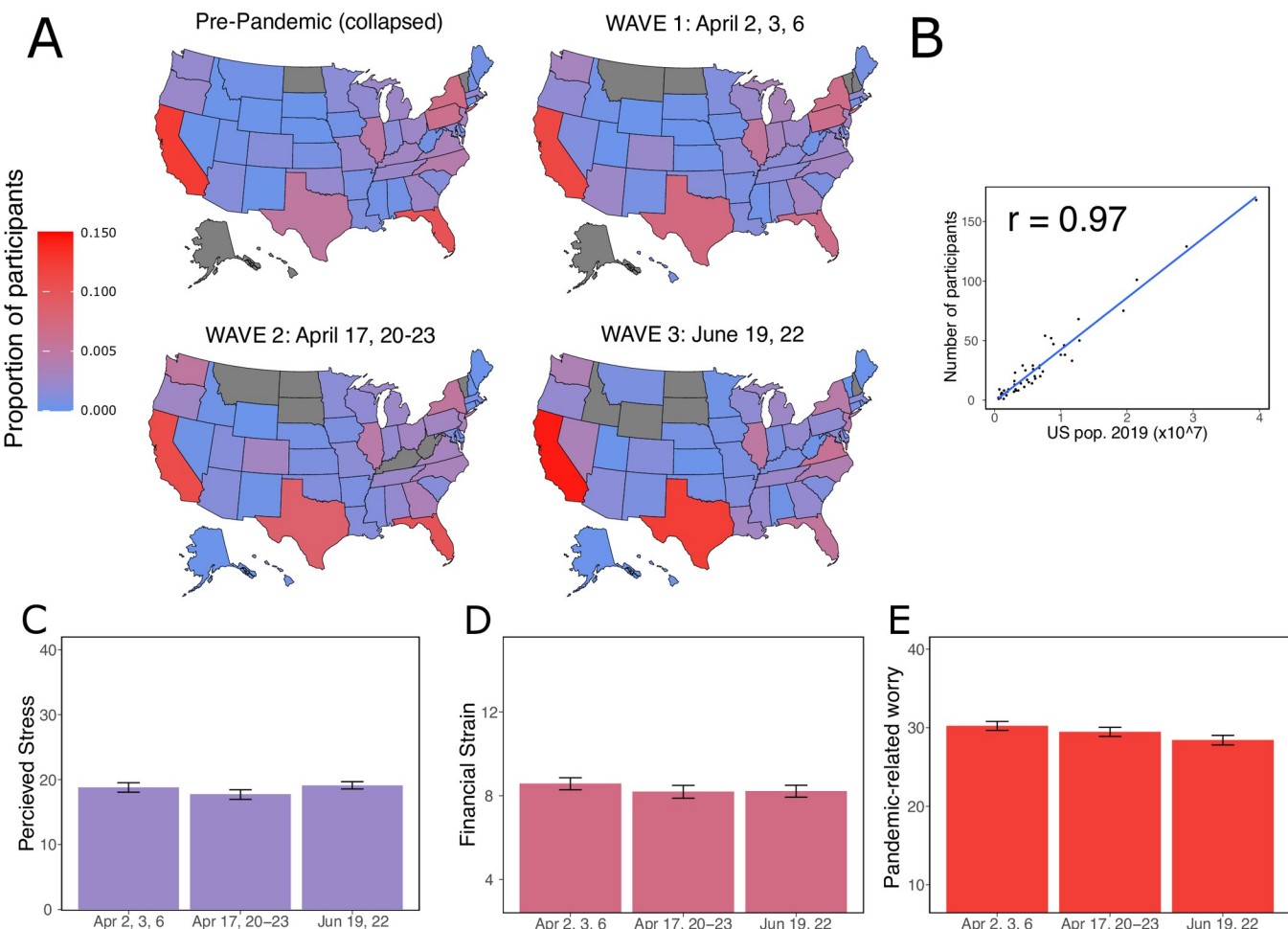

**Fig 1. Summary of participant responses across waves of data collection. A.** Map of the U.S. depicting the distribution of participants collected during the three samples. **B.** Correlation between the number of responses per state and the state population. **C.** Participants reported moderate levels of overall perceived stress which did not vary as a function of wave ($\beta$ = -0.321, CI = -0.841–0.198, p = .255); plotted here with 95% bootstrapped confidence intervals. **D.** Mean self-reported financial strain also decreased with pandemic progression (wave $\beta$ = -0.232, CI = -0.457—-0.007, p = .042); again, plotted with 95% bootstrapped confidence intervals. **E.** Mean self-reported coronavirus fear/worry (FCQ) decreased as a function of wave ($\beta$ = -1.188, CI = -1.64—-0.733, p < .001); again plotted with 95% bootstrapped confidence intervals.

the data collected—particularly that possible declines were not simply due to noisier responding (see Supplemental Material in S1 File). Crucially, exclusion from our sample was not found to covary with the key variable of interest—pandemic worry (see Supplemental Results in S1 File). Given the cross-sectional design of the study, it is essential to establish that the recruited samples are comparable in terms of demographic variables. To determine the similarity of the samples, we compared the three recruited samples both to each other and to the collapsed pre-pandemic samples on several demographic variables (see S15, S16 Tables in S1 File). Overall, participants had comparable ages across waves 1 ($M_{age}$ = 36.2, $SD_{age}$ = 9.87), 2 ($M_{age}$ = 36.5, $SD_{age}$ = 10.4,), and 3 ($M_{age}$ = 36.6, $SD_{age}$ = 10.4,), and in comparison to the collapsed pre-pandemic sample ($M_{age}$ = 37.8, $SD_{age}$ = 14.6, $F_{(3, 1753)}$ = 1.655, *p = 0.175*). In terms of reported gender, the third wave (females = 138, males = 327, other = 0) contained proportionally more male participants than the first wave (females = 172, males = 275, other = 4), second wave (females = 166, males = 287, other = 2), and collapsed pre-pandemic sample (females = 164, males = 224, other = 4). To control for demographic differences between

samples, we included age, gender, income, years on MTurk, and education level as covariates. We further controlled for pandemic-specific variables which may covary with performance: number of adults and children living at home given the stay-at-home orders, perceived risk for contracting COVID-19, and self-reported COVID symptoms [42].

## Procedure and materials

To assess the extent of the COVID-19 pandemic's impact on individuals' cognitive functioning and behaviour, we asked participants to complete a battery of 4 different tasks in a counterbalanced order. These tasks were identical to those used in our pre-pandemic samples in terms of stimuli, trial number and response-deadlines.

**Digit-symbol coding task.** To measure processing speed, participants were asked to complete a computerized version of the Digit-Symbol Coding task [43, 44]. During this task, participants are shown a static list of 9 digit-symbol pairs, which remain visible at the top of screen for the entirety of the task (see Fig 2A). On each trial, participants are asked to indicate whether the digit-symbol pair presented in the centre of the screen matches one of the 9 digit-symbol pairs depicted at the top of the screen. Yes/No responses were made using the left and right arrow keys, with the response-key mappings counterbalanced between participants. Following prior work [43], participants were asked to respond correctly to as many trials as they could within 90 seconds. In keeping with previous pre-pandemic samples collected in the lab, we implemented an attention check designed for online data collection where participants who did not achieve 70% accuracy on the task were asked to complete the task a second time (28% of participants in wave 1; 33% in wave 2; and 42% in wave 3). In this case, only data from the second run were analyzed [45]. We compared the current samples to the previously collected data [45].

**Task-switching paradigm.** To measure task-set shifting ability, we asked participants to complete a task switching paradigm [22, 46] in which participants were shown a square either on the top or bottom half of the screen (see Fig 3A). Depending on the position of the square, participants were asked to indicate the colour (i.e. orange or blue) or the pattern of the box (i.e. solid or stripped) using either the using the 'E' or 'I' buttons on the keyboard (e.g. blue = "E", orange = "I"; solid = "E", striped ="I") within the 1500ms time limit. Critically, both the position-task mappings, and the key-response mappings were counterbalanced between participants. Participants completed 80 trials, in which half switched between subtasks (e.g., from color to pattern) and the other half repeated the previous subtask (e.g., from color to color). These data were compared to the "preliminary phase" data of previously collected data [47].

**Dot Pattern Expectancy task (DPX).** We assessed proactive cognitive control by using the dot-pattern expectancy task (DPX) [48]. During this task, participants were first shown a dot pattern which served as the cue for 500ms (blue dot pattern; *A* or *B*), followed by a fixation cross for 2000ms (the delay period), and finally the probe stimulus for 500ms (white dot pattern; *X* or *Y*). Participants were instructed that a specific combination of cue and probe dot patterns would serve as the "target" stimulus and were told to respond using the "1" key when presented the "target" cue-probe pair—which we refer to as the AX trial—and respond with the "2" key with all other non-target combinations—AY, BX, BY (see Fig 4). Participants completed a total of 128 trials, where the majority were target trials (AX, 68.75%), to establish the prepotency of the "target" response. The remaining trials were equally distributed among the three other trial types (BX, BY, AY). Using participant's RTs, we calculated the proactive behavioural index (PBI) [49] which reflects the relative inference on AY and BX trials–calculated as $(RT_{AY}-RT_{BX})/(RT_{AY} + RT_{BX})$, where a larger PBI reflect more utilization of cue-based

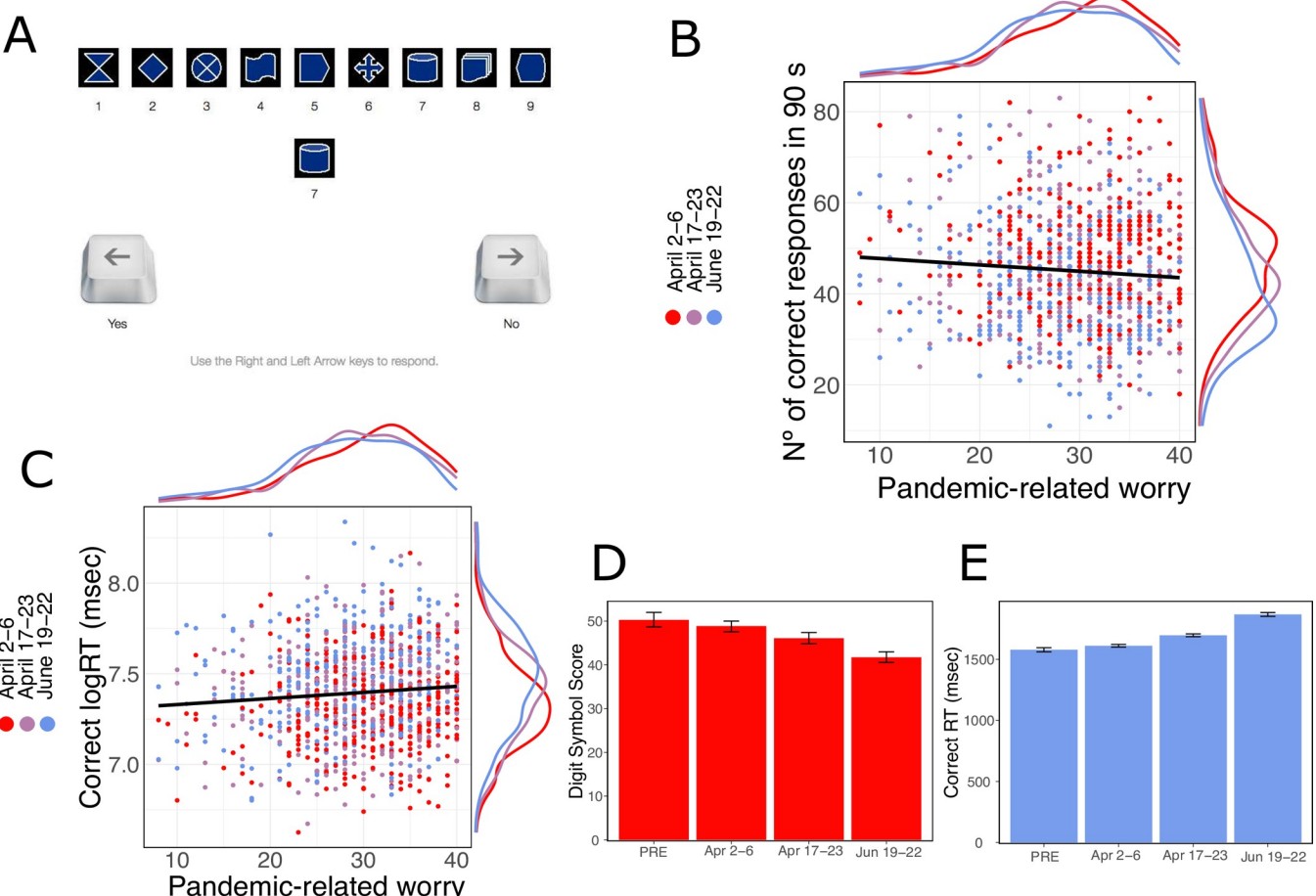

**Fig 2. Participants' performance on the digit-symbol coding task. A.** Screenshot of the Digit symbol-coding task. **B.** We observed a relationship between FCQ scores and Digit Symbol-Score (β = -0.940, CI = [-1.591, -0.289], p = .005); here plotted with the marginal distribution of the ordinate and abscissa **C.** We also observed a relationship between FCQ scores and log correct RTs (β = 0.019, CI = [0.005, 0.033], p = .009); again, plotted with the marginal distribution of the ordinate and abscissa. **D.** Processing Speed decreased in comparison to the pre-pandemic sample and by wave (p's < .05;); here plotted by wave with 95% bootstrapped confidence interval. **E.** Participants' response times increased in comparison to the pre-pandemic sample and by wave (p's < .05); again, plotted here with 95% bootstrapped confidence intervals.

contextual information (and less probe-driven behavior) [49, 50]. Note that our pre-pandemic DPX sample [51] lacked self-reported age or gender, and accordingly these variables they were not included in the linear model comparing across pre-pandemic and pandemic samples.

**Risky decision-making task.** Participants' risk preferences were assessed using a simple risky decision-making task in which they were presented with six practice trials and 120 binary choices between a risky and a certain option. For half of the choices, options were framed as losses (i.e., either -$200 or -$100) and the remaining half were framed as gains (i.e., either $200 or $100). All risky options were associated with both a chance of winning (or losing) a non-zero amount of hypothetical money and a chance of winning (or losing) nothing at all (see Fig 5). For the risky options, the probability of the non-zero outcome varied between likely (0.90, 0.95 or 0.99) or unlikely (0.10, 0.05, 0.01) outcomes [36]. All choices were between options of equal expected value (EV), except for twelve "catch" trials in which the expected value greatly favored an option (expected value = +/- 90), ensuring that participants understood the task (see S11 Table in S1 File for full list of stimuli used). Participants were asked to select an option within a 2500ms limit by responding with either the left or right arrow key. Data was

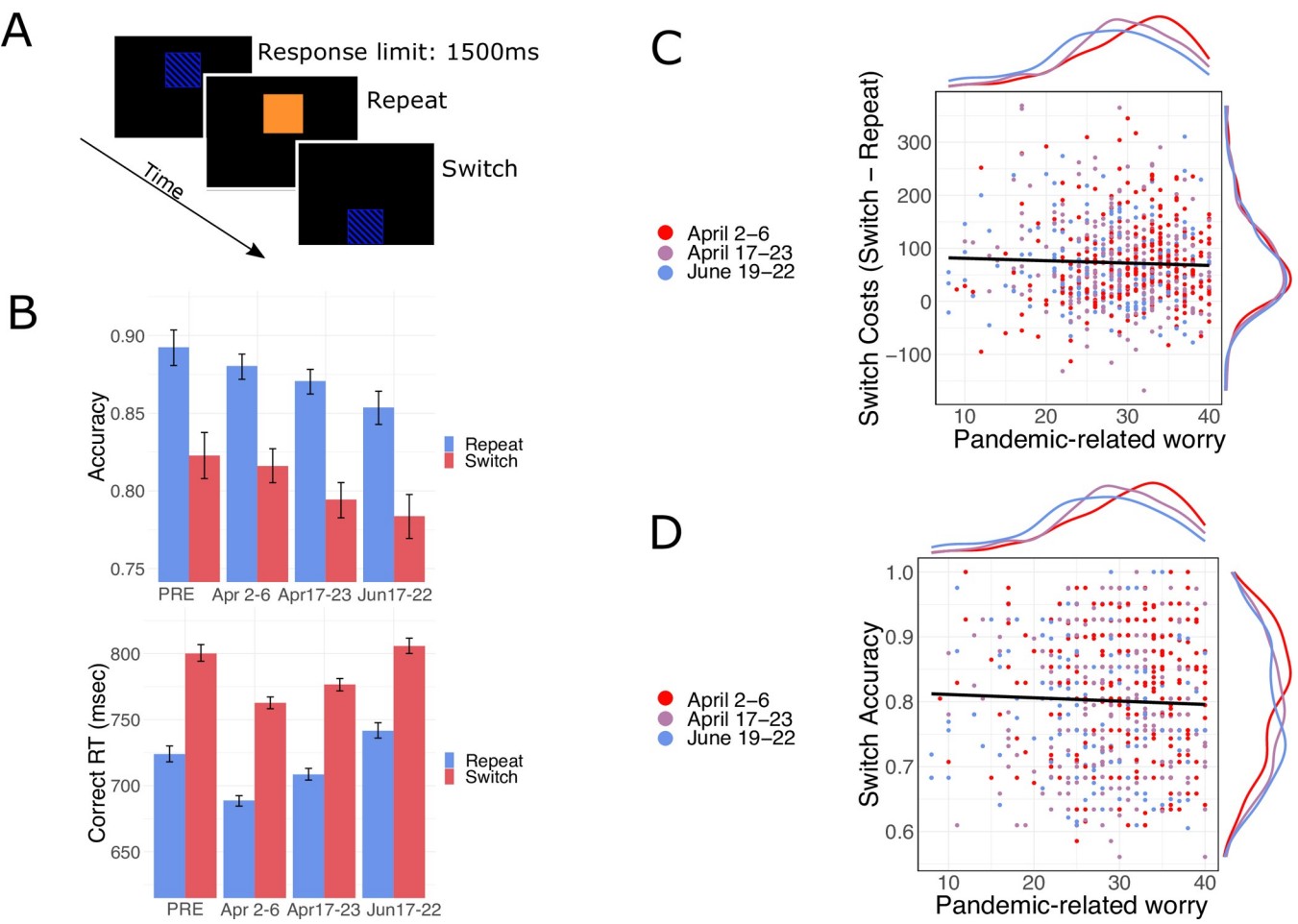

**Fig 3. Participants' performance on the Task-Switching paradigm. A.** Schematic of task-switching paradigm. **B.** Accuracy (top; p's < .05) but not response-times (bottom) decreased by wave (p's ≥ .05); variables plotted with 95% bootstrapped confidence interval. **C.** There was no observed relationship observed between FCQ scores and switch costs expressed in RT (β = -0.0005, CI = [-0.0142, 0.0132], p = .946); here, plotted with marginal distributions for the ordinate and abscissa and **D.** There was no observed relationship between FCQ scores and switch trial accuracy (β = -0.0385, CI = [-0.0865, 0.0096], p = .117); again, plotted with marginal distributions for the ordinate and abscissa.

compared to a pre-pandemic sample, previously collected, where participants completed the same risky decision-making task under two deadlines (2500ms and 3500ms) [45].

**Questionnaires.** First participants completed all four behavioural tasks in a counterbalanced order, then the PSS, FSS, and the FCQ questionnaire in a randomized order. Finally, participants were asked to complete a final questionnaire assessing demographic variables and rated their ability to focus.

*Fear of Coronavirus Questionnaire–(FCQ).* To assess anxiety related to the pandemic, we administered an 8-item questionnaire assessing beliefs and behaviours relating to the COVID-19 pandemic [28]. Participants were asked to rate on a 5-point scale, the extent to which they endorsed statements relating to fear of the virus (e.g., "I am very worried about the coronavirus outbreak.") or engaging in certain behaviours (e.g., "I am constantly following all news updates regarding the virus."). Scores on this questionnaire ranged from 8 to 40, where larger numbers reflect greater fear.

*Perceived Stress Scale.* The Perceived Stress Scale is a 10-item instrument which measures an individual's perceived level of stress over the last month [52]. Participants were asked to

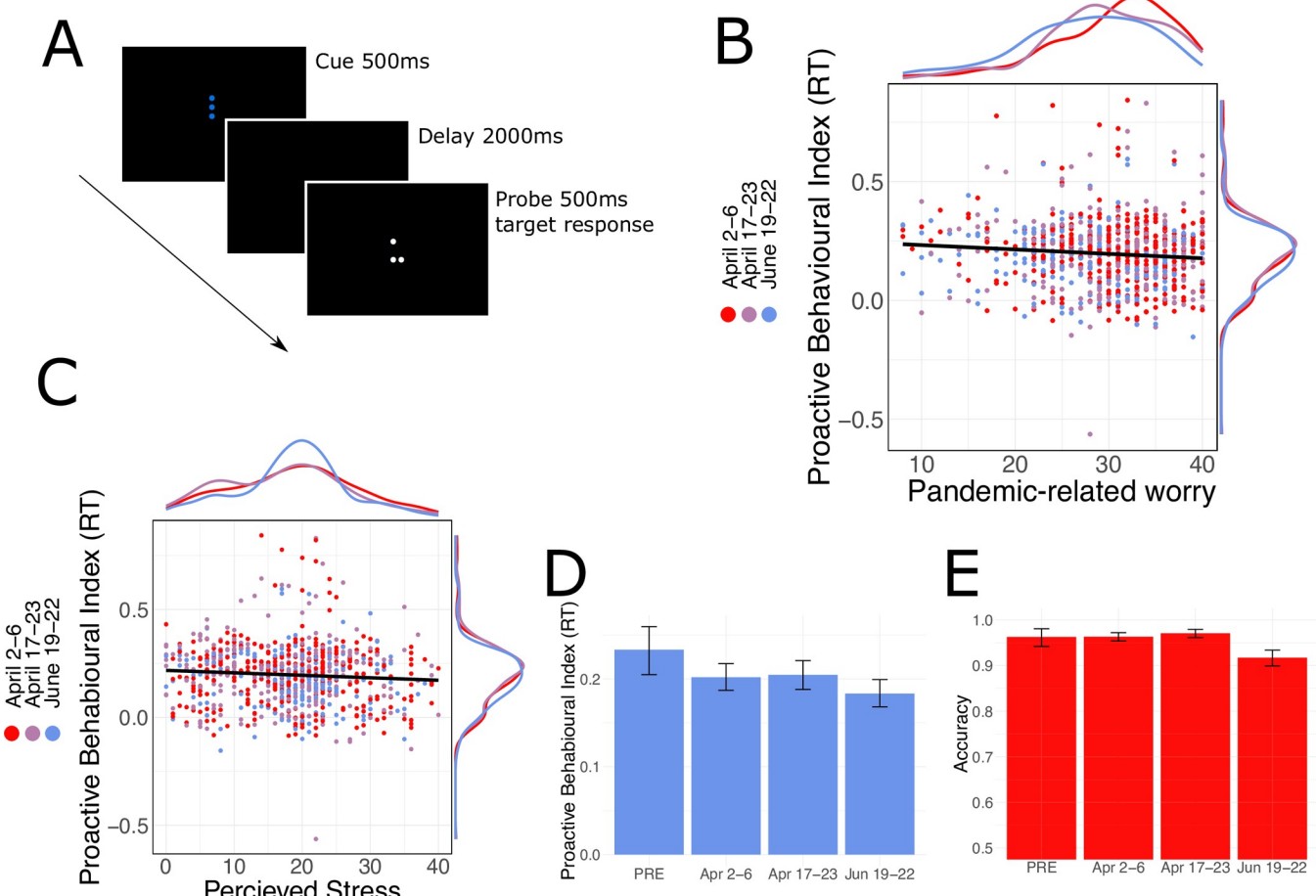

**Fig 4. Participants' performance on the dot pattern expectancy task. A.** Schematic of the dot pattern expectancy task where a specific combination of cue (shown in blue) and probe (shown in white) dots were deemed to be the "target" stimulus and associated with target responses, all other combinations required non-target responses. **B.** We observed a negative relationship between FCQ scores PBI (β = -0.0020, CI = [-0.0034, -0.0005], p = .010); here plotted with marginal distributions for the ordinate and abscissa. **C.** We observed a negative relationship between perceived stress scores and PBI (β = -0.0108, CI = [-0.0200 –-0.0015], p = .023); again, with marginal distributions plotted for the ordinate and abscissa. **D**. Proactive behavioural index (PBI) decreased in comparison to the pre-pandemic sample (β = -0.0353, CI = -[-0.0674, -0.0032], p = .031) but not with pandemic progression (p's > .20); error bars represent the 95% bootstrap confidence interval. **E.** Overall accuracy was not found to decrease as a function of pandemic progression (p>.05).

rate how often they felt overwhelmed (e.g., "In the last month, how often have you felt nervous and 'stressed'?") or in control (e.g., "In the last month, how often have you felt that you were on top of things?") on a scale from 1 "very often" to 5 "never". Responses to the reverse-coded questions were reversed-scored, such that larger scores reflect more stress, and range from 10 to 50.

*Financial Strain Scale (FSS)*. We used a 3-item measure of financial strain which asks participants to rate, on a 5 point scale, how much they endorse statements conveying financial hardships (e.g., "in the next two months, how much do you anticipate that you or your family will experience actual hardships such as inadequate housing, food, or medical attention?") [53]. Scores ranged from 3 to 15, where larger numbers reflect more strain.

*Self-reported ability to focus*. At the end of the cognitive tasks, participants were asked to rate their ability to focus during the experiment using a 9-item scale which assessed the degree of understanding (e.g., "all instructions were clear"), motivation (e.g., "I did my best on the task at hand"), and distraction (e.g., "I was distracted during the experiment") [27]. Ratings

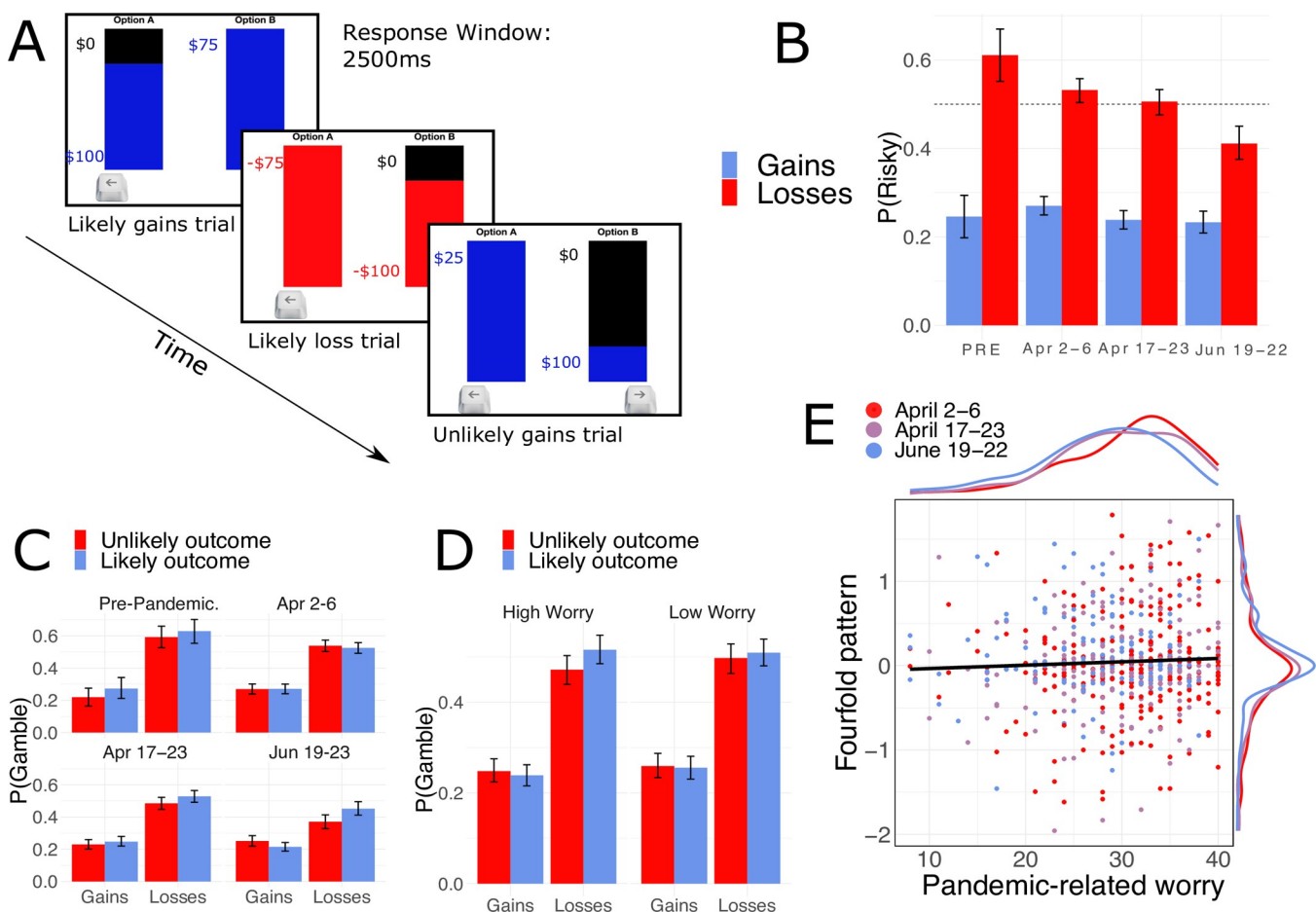

**Fig 5. Participants' performance on the risky decision-making task. A**. Schematic of the economic decision-making task used to assess risk preferences. **B.** Overall, participants were loss averse, as demonstrated by their sensitivity to the framing of problems as either losses or gains (β = -1.3200, CI = [-1.4430, -1.1969], p < .001) which also varied as a function of pandemic wave (β = 0.1949, CI = 0.0440, 0.3458], p = .011). **C.** Average proportion of risky choice by frame, outcome probability and sample. Participants in the third wave, compared to the first and the pre-pandemic sample, were more likely to distort outcome probability (p ≤ .001) **D.** Average proportion of risky choice by frame, outcome probability and fear of coronavirus questionnaire (FCQ) scores median split. **E.** We observed a positive relationship between fear of coronavirus and one's tendency to distort outcome probability (β = 0.1720, CI = [0.0929, 0.2511], p <0.001)—indexed as the magnitude of the fourfold pattern ((P(risky)$_{Likely\ Losses}$—P(risky)$_{Unlikely\ Losses}$)+ (P(risky)$_{Unlikely\ Losses}$—P(risky)$_{Likely\ Gains}$)); plotted here with marginal distributions for the ordinate and abscissa.

were taken on a 11-point scale, resulting in a range of possible scores from 0 to 90, where lower scores reflect less focus.

*Demographic variables.* In addition to the above measures, participants were asked to indicate several demographic variables to be included as covariates in the regressions. We asked participants to indicate their age, gender, highest level of education achieved, their annual household income, and the number of adults and children currently living with them. Additionally, we asked participants to specify if English was their first language, and if not, what age did they first start learning English. To account for the possibility that the pandemic could have attracted new people to Mturk, we asked participants to indicate how long have they been a worker on Mturk. Finally, participants were asked to indicate if they felt sick with any COVID-19 symptoms and to determine if they believed themselves to be at an elevated risk of contracting a more severe case of COVID-19 (i.e., "*Do you think you are at increased risk of experiencing a more severe case of COVID-19/coronavirus (i.e,. due to an underlying medical condition)?*") [54].

### Data analysis

To test the effect of worry on cognitive functioning, we ran separate hierarchical linear or logistic mixed-effect models for each task to assess the extent to which response times or response accuracy related to pandemic worry, controlling for several covariates (Age, Gender, income, highest education level, perceived risk for contracting COVID-19, total years on MTurk, and number of adults and children living in the same household). For the multi-level models, we included the within-participant trial-by-trial predictor variables as both random and fixed effects (i.e. task switch versus repetition, response congruency, decision frame losses versus gains, outcome probability likely versus unlikely). This allowed us to estimate group- and participant- level estimates for the effects in question (e.g., switch costs). We further estimated fixed-effects linear models to examine the relationship between singular, per-participant summary measures of task performance (i.e., total number of correct trials on the digit-symbol task, PBI) and both predictor variables (wave, worry, perceived, and financial stress) and subject-level covariates.

Beyond the main effects of interest (i.e., pandemic-worry), we also explored the overall effect of the pandemic on cognitive functioning. To this end, we ran separate linear or logistic mixed-effect models for each task where pandemic participants (all three waves collapsed) were compared to the pre-pandemic sample with a dummy coded variable (i.e. Pre-Pandemic: 0 vs Pandemic: 1). We also evaluated whether the individual measures of stress and anxiety changed as a function of pandemic progression (i.e., between waves) using simple linear regressions. As with the worry-analyses, we controlled for sample differences (i.e., gender and age) where possible; however, as these pre-pandemic and wave comparisons are cross-sectional, they should be taken as exploratory. All analyses were conducted using the *lme4* package (version 1.1–21) for the R programming language.

## Results

We collected US-based online samples across three waves corresponding to the early phase of the COVID-19 pandemic in North America: April 2nd, 3rd, and 6th, April 17th and 20th -23rd, and June 19th and 22nd, 2020 (see Fig 1A). Within this period, many of the U.S. states enacted strict regulations to mitigate the spread of the virus, including travel bans, limitations on gatherings, and stay-at-home directives, indicating that the pandemic was widespread at the time of data collection. Importantly, we also observed a strong relationship between the number of responses in each U.S. state and that state's population in 2019 ($r = 0.976$, p $< .0001$; collapsed over waves), suggesting that the geographical distribution of participants was consistent with the state-level population distribution (Fig 1B). We measured participants' performance on three cognitive tasks, an economic choice task, pandemic-related worry (e.g., subjective worry, looking up pandemic information) using the Fear of Coronavirus Questionnaire (FCQ) [28], and both financial stress (FSS) [53] and overall perceived stress (PSS) [52].

### Experienced fear, worry and stress

As depicted in Fig 1C and 1D, participants reported experiencing both moderate financial ($M_{FSS} = 8.35$; $SD_{FSS} = 3.22$) and overall perceived stress ($M_{PSS} = 18.63$; $SD_{PSS} = 7.42$) as measured by the FSS [53] and the PSS [52] respectively. Additionally, participants' self-reported pandemic worry—measured using the FCQ [28] ($M_{April} = 29.81$; $SD_{April} = 6.40$; $M_{June} = 28.42$; $SD_{June} = 6.73$)—matched the levels reported by other work assessing worry in an international sample during the same time period [55]. Relatedly, perceiving oneself as being at an elevated risk for contracting COVID-19 was associated with higher levels of pandemic-related worry ($\overline{\beta} = 2.3799$, 95% CI = [1.5289–3.2309], $p < 0.001$) [54]. Next, we compared pandemic worry,

perceived overall stress, and financial stress across waves (Fig 1E), finding that pandemic-related worry decreased linearly with wave ($\overline{\beta}$ = -1.188, CI = [-1.642, -0.733], $p< .001$) mirroring previous observations taken during the North American pandemic onset [55, 56]. Financial stress ($\overline{\beta}$ = -0.232, CI = [-0.457, -0.007], $p$ = .042), but not perceived stress levels were found to decrease as a function of pandemic progression (linear effect $\overline{\beta}$ = -0.321, CI = [-0.841, 0.198], $p$ = .225).

## Processing speed

We measured participants' ability to quickly process information [21], using the digit-symbol coding task, where participants were required to indicate, for as many stimuli as possible within 90 seconds, whether the digit-symbol pair displayed matched one of the pairs displayed in the digit-symbol key at the top of the screen [21] (Fig 2A). Interestingly, we observed that higher levels of pandemic-related worry—indexed by the FCQ—predicted lower performance (Fig 2B). Statistically, this predictive effect was significant, controlling for age, gender, overall stress, financial strain, perceived risk of contracting COVID-19, and several other demographic variables (FCQ $\overline{\beta}$ = -0.940, CI = [-1.591, -0.289], $p$ = .005; see S1 Table in S1 File). Additionally, we found that even while controlling for pandemic-related worry, perceived risk of contracting a severe case of COVID-19 was also predictive of slower overall processing speed ($\overline{\beta}$ = 0.0362, 95% CI = [0.0003–0.0722], $p$ = 0.048; see S1 Table in S1 File). We also examined whether the predictive effect of worry on digit-symbol performance could be explained by a decrease in response caution, as indexed by faster correct RTs [43]. Instead, we found that increased FCQ scores were associated with slower correct response times (RTs; $\overline{\beta}$ = 0.019, CI = [0.005, 0.033], $p$ = .009; see Fig 2C and S2 Table in S1 File) after controlling for the covariates listed above, indicating that the onus of worry's predictive effect was a general slowing of information processing.

Next, in an exploratory analysis, we compared performance across waves (Fig 2D) and with a pre-pandemic sample taken in the US in 2018–2019 (see S3 Table sample information in S1 File). We found that pandemic samples exhibited lower digit-symbol scores ($\overline{\beta}$ = -5.470, CI = [-7.640, -3.300], $p$ <0.001) and responded more slowly ($\overline{\beta}$ = 0.105, CI = [0.059, 0.151], $p$ <0.001) compared to the pre-pandemic sample, controlling for participants' age and gender. Across pandemic waves, we also observed a linear decline in processing speed as a function of pandemic progression ($\overline{\beta}$ = -1.120, CI = [-1.980, -0.259], $p$ = .011; see Fig 2D): we found digit-symbol scores were lower in the second ($\overline{\beta}$ = -2.295, CI = [-3.863, -0.728], $p$ = .004) and third waves ($\overline{\beta}$ = -2.124, CI = [-3.8492, -0.4000], $p$ = .0168; see S1 Table in S1 File), relative to the first wave, controlling for FCQ scores. This pattern was echoed in RTs, where participants were slower in the latter two waves (e.g., Waves 2 and 3) in comparison to the first wave ($p$'s < .05, see Fig 2E and S2 Table in S1 File). In short, the observed impairments in processing speed, both in association with pandemic-related worry and pandemic progression, hint that a worry-induced reduction in basic information processing ability may be a cognitive consequence of the pandemic.

**Task-switching.** We examined individual's ability to reconfigure mental processes in response to a change in goals, indexed by switch costs—the slowing of responses following switches between subtasks [22, 46]. In the task-switching paradigm, participants are required to respond to a stimulus based on a subtask that varied from trial-to-trial (see Fig 3A). On half of the trials, the required subtask (COLOR versus PATTERN) repeated, while the other half of trials entailed a switch to the other subtask, yielding "repeat" and "switch" trials respectively.

As is typical with task switching paradigms, we observed significantly slower RTs on switch trials compared to repeat trials ($\overline{\beta}$ = 0.1037, CI = [0.0930, 0.1144], $p$ <0.001, see Fig 3B and S4

Table in S1 File), reflecting switch RT costs [22]. We examined whether pandemic-related worry predicted task performance, but failed to observe a statistically meaningful relationship between FCQ scores and RTs ($\overline{\beta}$ = -0.0005, CI = [-0.0142, 0.0132], $p$ = 0.946; see S4 Table in S1 File), nor switch costs ($\overline{\beta}$ = -0.0015, CI = [-0.0080, 0.0050], $p$ = 0.647; see Fig 3C and S4 Table in S1 File), suggesting that worry may not have affected set-shifting ability.

Examining task-switching accuracy, we observed a typical decrease in response accuracy on task switches compared to task repetitions ($\overline{\beta}$ = -0.5719, CI = [-0.6531, -0.4907], $p$ <0.001; see S5 Table in S1 File) [22]. As depicted in Fig 3D, we failed to find a relationship between either pandemic-related worry (FCQ) and overall accuracy ($\overline{\beta}$ = -0.0385, CI = [-0.0865, 0.0096], $p$ = .117; See S5 Table in S1 File) or switch-costs expressed in terms of accuracy ($\overline{\beta}$ = 0.0066, CI = [-0.0415, 0.0547], $p$ = .788; see S5 Table in S1 File). Together, these results suggest that, unlike processing speed, general task-set shifting abilities—indexed by task switch costs—were unrelated to worry.

In terms of our exploratory analysis, we found that the pandemic samples were generally less accurate compared to the pre-pandemic sample ($\overline{\beta}$ = -0.2443, CI = [-0.3835, -0.1051], $p$ = 0.001; see Fig 3B and S6 Table in S1 File), controlling for age and gender. Overall, accuracy was lower in the second and third waves compared to the first ($p$'s < .05; see S5 Table in S1 File), after controlling for age, gender, perceived risk for contracting COVID-19 and worry (see S7 Table in S1 File for full list of covariates). However, we did not find any differences in either RTs ($\overline{\beta}$ = -0.0066, CI = [-0.0437, 0.0304], $p$ = .726; see S7 Table in S1 File), or switch RT costs ($\overline{\beta}$ = -0.0045, CI = [-0.0045, 0.0110], $p$ = .572) between the pandemic and pre-pandemic sample collected in 2019. Similarly, comparing across waves, we did not observe an effect of pandemic wave upon RTs when comparing between waves (all $p$'s $\geq$ .05; see S4 Table in S1 File).

**Proactive cognitive control.** We probed whether pandemic-related worry had any predictive bearing on proactive cognitive control, assessed with the Dot Pattern Expectancy task (DPX) [48]. In the DPX task (Fig 4A), a cue stimulus (blue dot pattern; *A* or *B*) is presented briefly, followed by a delay and a probe stimulus (white dot pattern; *X* or *Y*). Participants were instructed to make a 'target' response only to a valid cue-probe pair (denoted *AX*), and a 'non-target' response for all other invalid cue-probe pairs: *BX*, *AY*, and *BY*. Importantly, target (*AX*) trials are far more frequent than non-target trials (approximately 2/3 of trials), engendering a preparatory context triggered by the *A* cue, and a bias for target responses triggered by the *X* probe.

The relative performance on BX versus AY trials is used as a measure of proactive control, because the cue-driven expectancy set by the *B* context can be utilized to inhibit the incorrect, but prepotent target response to the *X* probe [57]. Accordingly, engaging in the use of proactive control supports performance on *BX* trials, but at the same time, can impair performance on non-target *AY* trials. Conversely, reactive control relies more on the probe, which is thought to impair performance on non-target *BX* trials but improve *AY* performance. Overall, DPX RTs and accuracies in the pandemic sample mirrored typically observed patterns [51, 58]: subjects made faster and more accurate responses to target *AX* trials and non-target *BY* trials compared to *AY* and BX (RT cue × probe interaction $F$ = 254.7, $p$ < .001; Accuracy interaction $F$ = 905.57, $p$ < .001; see S8 Table in S1 File).

We next examined whether pandemic-related worry predicted deficits in the proactive behavioural index (PBI) [49]—an RT-based measure of proactive control, calculated as $(RT_{AY} - RT_{BX}) / (RT_{AY} + RT_{BX})$—finding that individuals with higher FCQ scores had lower PBIs (Fig 4B; $\overline{\beta}$ = -0.0020, CI = [-0.0034, -0.0005], $p$ = 0.010; see S9 Table in S1 File), controlling for gender, age, perceived risk of contracting COVID-19, self-reported, and financial

stress (see S8 Table in S1 File for full list of covariates). That is, individuals reporting more pandemic-related worry had greater difficulty inhibiting stimulus-driven responding. Additionally, we observed a negative relationship between reported chronic stress (measured with the PSS) and PBIs ($\bar{\beta}$ = -0.0108, CI = [-0.0200 –-0.0015], $p$ = 0.023, see Fig 4C). In summary, these results suggest that pandemic-related worry is associated with deficits in proactive cognitive control.

Relative to a pre-pandemic sample collected in 2013, the pandemic sample exhibited significantly—albeit slightly—reduced PBIs ($\bar{\beta}$ = -0.0353, CI = -[-0.0674, -0.0032], $p$ = 0.031; Fig 4B), but no pandemic progression effect (linear effect of wave: $\bar{\beta}$ = -0.0075, CI = [-0.0204, 0.0054], $p$ = .254; see Fig 4D) nor any significant PBI differences between the first wave and the latter two waves (*p's > 0.20* & S8 Table in S1 File). Importantly, overall response accuracy did not vary as a function of wave or FCQ (*p's* > .05; see Fig 4E and S10 Table in S1 File).

**Risky decision-making.** Finally, we examined how pandemic-related worry relates to risk preferences as a function of effect-coded variables of gain-loss decision framing [59], outcome probability, and their possible interactions [36]. To do this, we measured risk preferences using a classic economic choice task (Fig 5A), wherein participants made a series of hypothetical choices between a 'certain' option e.g., a sure win of $75, and a 'risky' option e.g. a 25% chance of winning $0 and a 75% chance of winning $100 (see S11 Table in S1 File for full list of stimuli used). These choices varied both in terms of the decision frame (i.e., losses versus gain outcomes) and the probability of the non-zero risky outcome, which varied from unlikely (1–10%) to likely (90–99%).

Overall, we observed participants were sensitive to the decision frame in accordance with the framing effect [35, 36]: participants' choices were risk-averse for gains and risk-seeking for losses (gain/loss frame $\bar{\beta}$ = -1.3200, CI = [-1.4430 –-1.1969], $p$ <0.001; see S12 Table in S1 File and Fig 5B). We next examined if pandemic-related worry predicted risk preferences, finding that FCQ did not relate to individuals' overall risk-taking (FCQ main effect; $\bar{\beta}$ = - 0.0450, CI = [-0.1250, 0.0350], $p$ = .270; see S12 Table in S1 File), nor susceptibility to the framing effect (FCQ x Frame interaction; $\bar{\beta}$ = 0.0226, CI = [-0.0977, 0.1429], $p$ = 0.713; see S12 Table in S1 File). Participants were also sensitive to described risk level (Frame x Probability interaction $\bar{\beta}$ = 0.2560, CI = [0.1744, 0.3375], $p$ < .001; see Fig 2C), in accordance with the 'Fourfold pattern' of risk preferences: choices were relatively more risk-seeking in situations of unlikely gains and likely losses compared to situations of likely gains and unlikely losses [36]. This pattern of preference is thought to arise from the underweighting of likely probabilities and overweighting of unlikely probabilities. Individuals reporting greater pandemic-related worry appeared more sensitive to described risk level (Fig 5D): FCQ scores predicted participants' tendency to exhibit this fourfold pattern in choice (visualized continuously in Fig 5E) as indicated by a significant three-way interaction between frame, outcome probability and worry ($\bar{\beta}$ = 0.1720, CI = [0.0929, 0.2511], $p$ <0.001; see S12 Table in S1 File). As with the analysis of cognitive task performance, this relationship between sensitivity to outcome probabilities and individual worry remained after controlling for demographic variables, and perceived risk of contracting COVID-19 (see S12 Table in S1 File for full list of covariates).

In an exploratory analysis, we compared the 2019 pre-pandemic sample to the pandemic samples and observed a decrease in overall risk-taking (Sample main effect $\bar{\beta}$ = -0.3490, CI = [-0.6417, -0.0563], $p$ = .019; see S13 Table in S1 File) and a decrease in sensitivity to gain/loss framing (Frame x Sample interaction $\bar{\beta}$ = 0.7465, CI = [0.3242, 1.1688], $p$ = .001; see S13 Table in S1 File) after controlling for age and gender. Participants in the pandemic sample, compared to the pre-pandemic sample, also exhibited increased sensitivity to described risk level, i.e., greater expression of the fourfold pattern described above (Fig 5C), which was

supported statistically by an interaction between frame, outcome probability, and sample (pre-pandemic versus pandemic; $\overline{\beta}$ = 0.4723, CI = [0.1925, 0.7522], $p$ = .001, see S13 Table in S1 File). We did not observe an effect of pandemic progression on overall risk preferences (linear effect of wave $\overline{\beta}$ = -0.0884, CI = [-0.1819, 0.0123], $p$ = .085), but we did find that overall sensitivity to the gain/loss decision frame decreased as a function of wave (linear effect of wave x Frame interaction $\overline{\beta}$ = 0.1949, CI = [0.0440, 0.3458], $p$ = .011—participants in the third wave were found to be less sensitive to decision frame when compared to the first wave (Wave x Frame interaction $\overline{\beta}$ = 0.4281, CI = [0.1256–0.7306], $p$ = .006; see S14 Table in S1 File). We also found participants in the third wave showed greater expression of the fourfold pattern in comparison to the first wave (Wave x Frame x Probability interaction $\overline{\beta}$ = 0.6322, CI = [0.4278–0.8366], $p$ < .001, see S14 Table in S1 File). In summary, we found that both pandemic-related worry and pandemic progression were associated with a greater sensitivity to described risk level, but only pandemic progression impacted framing effects.

## Discussion

Here, we sought to delineate the cognitive and behavioural consequences of the COVID-19 global pandemic in a representative US sample. Consistent with our predictions, we observed a marked decrease in individual's executive control as a function of individual differences in experienced fear/worry. Together, the results outlined above buttress previously observed anxiety-induced impairments in executive functioning [13, 19, 60], and extend this literature by demonstrating a decline in executive control in response to the threats posed by the COVID-19 pandemic [16, 17]—a naturalistic stressor.

Previous accounts of anxiety posit that excessive worry impairs executive functioning by displacing cognitive resources (i.e., working memory) necessary for successful goal-directed control [18]. Supporting this view, we found that higher levels of pandemic-related worry predicted impairments in proactive cognitive control and in information processing speed but not in task-switching ability. We believe that this uneven effect of worry across cognitive tasks reflects the extent to which each task relies on working memory resources. On this view, we found that self-reported coronavirus anxiety measured by FCQ was associated with reduced use of goal-driven (i.e. proactive) control [19, 61, 62]—which is thought to rely more strongly on working-memory capacity than reactive control [25, 26]. Similarly, declines in processing speed, like those observed here, are thought to reflect both the slowing of basic mental operations, and limitations to the amount of simultaneously available information [21]. However, we failed to find a relationship between pandemic worry and task-switching ability. This is perhaps not surprising considering that studies of the effects of negative affective states on task-switching ability have revealed mixed findings [63, 64]. Critically, working-memory demands are thought to play a key role in anxiety-induced impairments in set shifting ability, as both task complexity [65], and response preparation [66] have been found to mediate the predictive effects of anxiety on task-switching. Thus, it is possible that the chosen task switching paradigm did not sufficiently tax working memory to observe a worry-induced decline in performance, as task complexity and response preparation were not manipulated. Dovetailing with these findings, we found that processing speed was related to performance on both the task-switching and proactive control tasks, but proactive control and task-switching performance did not appear to relate to one another (see S17 Table in S1 File). Together, these results highlight how executive functioning reflects a related, yet distinct set of cognitive abilities [24], and suggest that the effects of worry are not of equal magnitude across domains.

Whether poorer cognitive control is a function of, or a consequence of anxiety remains unclear, as previous work suggests that worrying interferes with cognitive ability [67], and that

lower cognitive ability contributes to worrying through an inability to inhibit negative thoughts [68]. Furthermore, other work has proposed that engaging in demanding cognitive tasks can reduce anxiety [69, 70], highlighting the hypothesized shared resources between worrying and goal-directed behaviour. Critically, as anxiety-related deficits in task performance did not generalize to all tasks, we believe it is unlikely that our results could be exclusively explained by exertion-induced reductions in anxiety. However, future work is needed to disentangle these possibilities.

In the context of a pandemic, individuals' risk assessments are equally crucial as they predict compliance with social-distancing measures [29]. Extant accounts suggest that anxiety shapes decision-making via an increased tendency to interpret uncertain information more negatively, though there are some inconsistencies [37]. Anxious individuals are thought to overestimate the likelihood of negative outcomes [30–32], leading to greater risk aversion [20, 33, 34]. However, we failed to find any evidence that greater worry was associated with risk aversion, nor loss aversion. Instead, we found that pandemic worry predicted individuals' tendency to distort described risk levels: underweighting likely probabilities and overweighting unlikely probabilities regardless or valence [36]. Perhaps, this disparity across studies in the observed effect of anxiety on choice reflects the use of different strategies to manage risk and/or worry [71].

One possible explanation for anxious individuals' greater sensitivity to described risk is an increase in information seeking, resulting in greater exposure to discussions of risk, which are the current focus of media coverage. On this view, intolerance of uncertainty has been associated both with greater pandemic worry [55], and greater information seeking—particularly when the uncertainty is related to one's health [72, 73]. Indeed, among individuals experiencing elevated levels of anxiety, there have been increases in searches for pandemic-related information [74]. While the risk attitudes measured here were unrelated to one's health, we believe this increased contemplation of risks, particularly for those experiencing anxiety, reflects an overall greater saliency of described risks on behaviour.

We also explored whether the pandemic had a general effect on task performance. We found that, in comparison to our pre-pandemic samples, those tested during the pandemic seem to exhibit slower processing speed [21, 43, 44], decreased task-switching accuracy [22, 46], less proactive control [49, 50], and greater sensitivity to described risk [36]. We further probed the effect of longer exposure to the stressor i.e., pandemic progression, controlling for worry, perceived risk of contracting COVID-19, financial and overall stress. While proactive control remained stable, we found that participants in the third wave, compared to the first, had slower processing speed, lower task-switching accuracy, and were more sensitive to risk. Given that we are comparing different groups of participants who were captured at different points in time, we can only speculate as to the mechanisms underlying the decline in cognitive performance across this 2-month period in the early stages of the pandemic. In keeping with other research [55, 56], we observed a decrease in the level of worry over time whereas task performance did not significantly change—suggesting there may be other factors contributing to impaired executive functioning. For example, the persistently impaired task performance may reflect the impact of prolonged (i.e. chronic) exposure to stress or a shift in the sources of worry. Supporting this view, we found that self-reported perceived stress remained stable across the waves of data collection. However, despite controlling for possible demographic differences between samples (e.g., age, gender, years on MTurk; see Supplemental Materials in S1 File), we cannot rule out the possibility that our groups (collected both in years prior to the pandemic and throughout) sampled different populations, as the pandemic was associated with a marked increase in financial hardships and remote work. Thus, the sample differences discussed here should be interpreted as exploratory and more work is needed to understand whether these wave effects reflect meaningful differences.

We believe the results outlined here have broader implications for human behaviour, as intact executive control is a cornerstone of healthy daily living (e.g., economic productivity), workplace performance [75], and the delay of gratification [76]. For instance, though we did not measure current compliance with COVID-19-related measures, it is possible that the behavioral repertoires examined here—executive functioning and risky decision-making—and the affective states (i.e., worry) of an individual might play a role in that individuals' understanding and compliance with public health measures. Indeed, recent work has observed a positive relationship between one's decision to comply with government sanctioned social-distancing directives and working-memory [77] as well as perceived (but not actual) pandemic-related risk [54]. Mirroring these findings, we found individuals who perceived themselves at higher risk for contracting COVID-19 also experienced greater pandemic-worry, and slower information processing (see S1 Table in S1 File). Given the putative role of coronavirus anxiety in successful goal-directed behaviour, it is possible that an individual's worry level plays a key role in the tendency to comply with public health measures: anxiety is associated both with worse working-memory and higher perceived risk for contracting COVID-19. One possibility is that pandemic-related worry could decrease compliance via putative changes in cognitive performance; another possibility is that worry could lead to increased compliance merely as a response to higher perceived risk of becoming sick, i.e., independently of the effects of worry on cognition. Given our cross-sectional design, our results cannot shed light on the possible directionality of these effects. Future research should examine the underlying factors contributing to cognitive function and decision-making under high stress to better understand the relationship between affective and cognitive states, and the impact of these states on real-world behaviours. This is of particular interest considering the declining adherence to government directives and the far-from-universal acceptance of vaccination [6, 7]. Our results also highlight the importance of considering the individual (i.e., worry) and the circumstantial (i.e., the pandemic) factors that lie outside of the experimenter's control but may contribute to measured behaviour—hereafter it may be unreasonable to assume that human participants enter experiments in a neutral state [78]. Going forward, we believe that future research should consider these individual differences when interpreting any behaviour relying on executive functioning—particularly for research conducted during, and immediately following the pandemic.

## Supporting information

**S1 File.**
(DOCX)

## Author Contributions

**Conceptualization:** Kevin da Silva Castanheira, Madeleine Sharp, A. Ross Otto.

**Data curation:** Kevin da Silva Castanheira.

**Formal analysis:** Kevin da Silva Castanheira.

**Funding acquisition:** Madeleine Sharp, A. Ross Otto.

**Investigation:** Kevin da Silva Castanheira.

**Methodology:** Kevin da Silva Castanheira, Madeleine Sharp, A. Ross Otto.

**Project administration:** Kevin da Silva Castanheira.

**Software:** Kevin da Silva Castanheira, A. Ross Otto.

**Supervision:** Madeleine Sharp, A. Ross Otto.

**Visualization:** Kevin da Silva Castanheira.

**Writing – original draft:** Kevin da Silva Castanheira.

**Writing – review & editing:** Kevin da Silva Castanheira, Madeleine Sharp, A. Ross Otto.

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
