## [Decision Letter · Decision Letter 0]

30 Sep 2021

PONE-D-21-23700

The impact of pandemic-related worry on cognitive functioning and risk-taking

PLOS ONE

Dear Dr. da Silva Castanheira,

Thank you for submitting your manuscript to PLOS ONE. After careful consideration, we feel that it has merit but does not fully meet PLOS ONE’s publication criteria as it currently stands. Therefore, we invite you to submit a revised version of the manuscript that addresses the points raised during the review process.

Your manuscript has been reviewed by two expert reviewers. Although both reviewers noted several strengths of the study, they also raised many important concerns that should be addressed in a revision. Both reviewers recommended adding additional details regarding the methods. In addition, Reviewer 1 raised several important points regarding the interpretation of the results, and Reviewer 2 identified a few key issues with the methods and analyses. Addressing these issues will strengthen the manuscript. 

We look forward to receiving your revised manuscript.

Kind regards,

David V. Smith, Ph.D.

Academic Editor

PLOS ONE

Additional Editor Comments (if provided):

Reviewers' comments:

Reviewer's Responses to Questions

**Comments to the Author**

1. Is the manuscript technically sound, and do the data support the conclusions?

Reviewer #1: Yes

Reviewer #2: Yes

2. Has the statistical analysis been performed appropriately and rigorously? 

Reviewer #1: Yes

Reviewer #2: Yes

3. Have the authors made all data underlying the findings in their manuscript fully available?

Reviewer #1: Yes

Reviewer #2: Yes

4. Is the manuscript presented in an intelligible fashion and written in standard English?

Reviewer #1: Yes

Reviewer #2: Yes

5. Review Comments to the Author

Reviewer #1: In this manuscript, the authors present results of a cognitive battery (processing speed, task switching, proactive vs. reactive cognitive control, and economic decision-making tasks) collected online in 3 waves between April-June 2020 (shortly after the COVID-19 pandemic began). They report that with increasing self-reported COVID worry, processing speed slows and proactive cognitive control (in terms of RT) decreases, and sensitivity to risky decision outcome probabilities increases. Task switching and decision risk/loss aversion did not significantly vary with COVID worry. Exploratory comparisons with pre-pandemic cognitive performance on these tasks suggests that cognitive performance was poorer and risk taking was reduced in the pandemic sample relative to pre-pandemic.

This is a valuable investigation on a timely issue, given the pervasiveness of stress and anxiety related to COVID-19 and the implications of associated changes in cognitive and decision performance. Overall I thought the paper was relatively sound; however, some additional information could have been provided and implications of the data explored further.

I was curious about how to interpret differences in the data between the waves collected. Specifically, it appears that pandemic-related worry decreased from Wave 1 to Wave 3, and some aspects of cognitive performance (i.e., processing speed) also declined from Wave 1 to 3, despite there also being a negative relationship between pandemic worry and processing speed. What additional sources of variance might have contributed to the performance decline? Should this be interpreted as noise?

I thought more information could be provided regarding the cognitive tasks as well as relationships in performance between them. For example, when describing the cognitive tasks, I was curious how long the task blocks of task switching, DPX, and decision performance were, and how this compared to the pre-pandemic data that they were compared against. I also was curious about correlations between performance and relationship with pandemic worry on different tasks (i.e., slowing on both the processing speed and DPX task) and whether examining for such correlations might help clarify why some metrics of executive function appeared to be modulated by pandemic worry and others were not.

I also was curious about the extent to which perceived risk for contracting COVID-19 (treated as a covariate) was related to COVID worry and whether this metric independently accounted for variance in performance. Sinclair et al. 2021, in PNAS, might also be relevant here.

Given that the paper framed COVID-related changes in cognitive performance in terms of understanding and complying with public health measures, I thought the authors could have discussed current public failures to comply with such measures in a bit more depth and nuance. Are the authors thinking of cognitive performance as a primary driver of such compliance, or other factors as well? Perhaps the observed decline in COVID worry that they observed over time continued past June 2020, and can be thought of as an affective factor contributing to failures in public compliance to COVID safety measures. I thought the paper could do a better job discussing potential contributions of such factors to public behavior as well as making clear the limitations of the present data and what kinds of contextual factors future studies might want to consider when examining cognitive performance under duress.

Reviewer #2: This research examined the relationship between Covid-19 related worry, working memory, and risk-taking behavior in an online sample. In general, I found the paper quite enjoyable — the abstract is exceptionally well-written. I think this paper would merit publication with minor revisions.

1. The primary thing I would like to see in a revised manuscript would be the correlation of the cognitive tasks with one another. This seems important particularly in light of the claims in the discussion that they perhaps differentially rely on working memory.

2. There is some use of acronyms without specifying what they are to the reader (FCQ; AX-CPT). I inferred, but it would be helpful to make it more explicit.

3. I’m confused about the analysis outlined on p. 10. I thought that covid related worry was predicting cognitive function, but it sounds like those were used as predictor variables, given that they are described as fixed and random effects?

4. Why was the Digit-Symbol Coding task run twice? I understand its due to accuracy, but aren’t they worried about practice effects? How many participants fell into this category? Were there similar accuracy constraints on the other tasks (it seems like no, why not?)

5. The claim that worry serves as a mediator to working memory disruption and correspondingly, obeying government sanctions is a big one. I would back off of that language, especially given that the link between worry, working memory, and Covid-19 related behavior were not tested in the scope of this paper.

6. There are a number of typos/omissions in the paper, and I would suggest a careful read before the next revision. I caught a few, listed below:

- “Dependant” is a typo on p. 10

- Figure 1, the caption is missing another variable name when it says “overall perceived stress did not vary as a function of”

- Figure 2E is missing the bars

- Figure 5e, the x axis is “Pandemic-related worry”

- Pg 19 is missing an of in “the effects negative stress on task-switching ability”

6. PLOS authors have the option to publish the peer review history of their article (what does this mean?). If published, this will include your full peer review and any attached files.

Reviewer #1: No

Reviewer #2: No

---

## [Author Response · Author response to Decision Letter 0]

18 Oct 2021

Reviewer #1: 

In this manuscript, the authors present results of a cognitive battery (processing speed, task switching, proactive vs. reactive cognitive control, and economic decision-making tasks) collected online in 3 waves between April-June 2020 (shortly after the COVID-19 pandemic began). They report that with increasing self-reported COVID worry, processing speed slows and proactive cognitive control (in terms of RT) decreases, and sensitivity to risky decision outcome probabilities increases. Task switching and decision risk/loss aversion did not significantly vary with COVID worry. Exploratory comparisons with pre-pandemic cognitive performance on these tasks suggests that cognitive performance was poorer and risk taking was reduced in the pandemic sample relative to pre-pandemic.

This is a valuable investigation on a timely issue, given the pervasiveness of stress and anxiety related to COVID-19 and the implications of associated changes in cognitive and decision performance. Overall I thought the paper was relatively sound; however, some additional information could have been provided and implications of the data explored further.

We appreciate the reviewer’s overall positive evaluation of our manuscript and thank the reviewer for their constructive comments on the clarity of the submission.

I was curious about how to interpret differences in the data between the waves collected. Specifically, it appears that pandemic-related worry decreased from Wave 1 to Wave 3, and some aspects of cognitive performance (i.e., processing speed) also declined from Wave 1 to 3, despite there also being a negative relationship between pandemic worry and processing speed. What additional sources of variance might have contributed to the performance decline? Should this be interpreted as noise?

We share the reviewer’s curiosity over the decreases in task performance over waves (e.g., processing speed) without a corresponding increase in self-reported subjective worry indexed by FCQ scores. In the discussion, we now provide more speculation as to the source of this variation. While it is possible that these differences reflect a meaningful change between waves (e.g., prolonged exposure to stress), it is also possible—as the reviewer points out—that these differences may reflect noise. Critically, given the cross-sectional nature of our study, we discuss the possibility that these wave differences in both performance and self-report anxiety reflect demographic differences in the samples associated with changes to the pool of MTurk participants—particularly as the pandemic was associated with a marked increase in financial hardship and work from home.

Page 29:

Given that we are comparing different groups of participants who were captured at different points in time, we can only speculate as to the mechanisms underlying the decline in cognitive performance across this 2-month period in the early stages of the pandemic. In keeping with other research [54,55], we observed a decrease in the level of worry over time whereas task performance did not significantly change—suggesting there may be other factors contributing to impaired executive functioning. For example, the persistently impaired task performance may reflect the impact of prolonged (i.e. chronic) exposure to stress or a shift in the sources of worry. Supporting this view, we found that self-reported perceived stress remained stable across the waves of data collection. However, despite controlling for possible demographic differences between samples (e.g., age, gender, years on AMT; see Supplemental Materials), we cannot rule out the possibility that our groups (collected both in years prior to the pandemic and throughout) sampled different populations, as the pandemic was associated with a marked increase in financial hardships and remote work. Thus, the sample differences discussed here should be interpreted as exploratory as more work is needed to understand whether these wave effects reflect meaningful differences.

I thought more information could be provided regarding the cognitive tasks as well as relationships in performance between them. For example, when describing the cognitive tasks, I was curious how long the task blocks of task switching, DPX, and decision performance were, and how this compared to the pre-pandemic data that they were compared against. I also was curious about correlations between performance and relationship with pandemic worry on different tasks (i.e., slowing on both the processing speed and DPX task) and whether examining for such correlations might help clarify why some metrics of executive function appeared to be modulated by pandemic worry and others were not.

The reviewer’s point on the clarity of our methods is well taken. We have made edits throughout the methods section to improve the clarity and transparency of the procedures used.

With respect to block length, all the tasks used in the current experiments were identical to those used in the pre-pandemic samples, both in terms of stimuli and duration (trial number, response-deadlines etc.). We thank the reviewer for pointing out this ambiguity and have clarified this point in our revision 

Page 8: 

These tasks were identical to those used in our pre-pandemic samples in terms of stimuli, trial number and response-deadlines.

Finally, with respect to correlations, we share the reviewers’ interest in the relationship among the various executive functioning tasks used and the observed pattern—FCQ predicted slower processing speed and proactive control but not task switching. As suggested, we opted to compute simple Pearson correlations among different metrics of task performance: digit symbol scores computed as the total number of correct responses within 90 seconds, PBI computed as the relative degree of interference on trials recruiting proactive vs reactive control (RTAY – RTBX)/(RTAY + RTBX), and the empirical bayes estimates of switch costs estimated using the linear hierarchical regression predicting log RTs. As seen in the correlation matrix below, we observe a statistically significant relationship between digit symbol scores and both proactive control, and task-switching costs. However, reliance on proactive control was not related to performance on the task-switching paradigm. As the reviewer intuited, this corroborates our results, while processing speed seems to be a more fundamental aspect of cognition—being related to performance on the other two tasks—proactive cognitive control and task-switching performance seem to index disparate aspects of executive functioning (Miyake et al., 2000). While these correlations do not provide direct evidence, they are aligned with the purported effects of worry on working memory capacity outlined in the discussion. 

 Switch Costs Digit Symbol Score PBI

Switch Costs 

Digit Symbol Score 0.112* 

PBI 0.002 0.124* 

Computed correlation used pearson-method with listwise-deletion.

* p < .05

Accordingly, we have updated the discussion section of the paper to reflect this new analysis. We thank the reviewer for their insightful comment, which has strengthened the overall argument of the submission.

Page 27:

However, we failed to find a relationship between pandemic worry and task-switching ability. This is perhaps not surprising considering that studies of the effects of negative affective states on task-switching ability have revealed mixed findings [62,63]. Critically, working-memory demands are thought to play a key role in anxiety-induced impairments in set shifting ability, as both task complexity [64], and response preparation [65] have been found to mediate the predictive effects of anxiety on task-switching. Thus, it is possible that the chosen task switching paradigm did not sufficiently tax working memory to observe a worry-induced decline in performance, as task complexity and response preparation were not manipulated. Dovetailing with these findings, here we found that processing speed was related to performance on both the task-switching and proactive control tasks, but proactive control and task-switching performance did not appear to relate to one another (see S17 Table). Together, these results highlight how executive functioning reflects a related, yet distinct set of cognitive abilities [23], and suggest that the effects of worry are not of equal magnitude across domains.

I also was curious about the extent to which perceived risk for contracting COVID-19 (treated as a covariate) was related to COVID worry and whether this metric independently accounted for variance in performance. Sinclair et al. 2021, in PNAS, might also be relevant here.

We share the reviewer’s intuition that perceived risk of contracting COVID-19 would contribute to task performance. While our previous submission had already controlled for perceived risk of contracting severe COVID-19, we recognize that these results were not presented clearly and with due emphasis. We have updated the manuscript accordingly. 

In our study, participants rated whether they perceived themselves at being of elevated risk for contracting a severe case of COVID-19 (i.e., “Do you think you are at increased risk of experiencing a more severe case of COVID-19/coronavirus (i.e,. due to an underlying medical condition)?“). This variable was then effects-coded (high vs low risk) based on whether participants did or did not endorse the statement, and entered into all appropriate regressions. Perceiving oneself as being at an elevated risk for contracting COVID-19 was associated with higher levels of pandemic-related worry (i.e., FCQ scores, B =2.3799, 95% CI =[1.5289 – 3.2309], p <0.001). In terms of task performance, beyond self-reported worry, those who perceived themselves to be at an elevated risk, were slower (B= 0.0362, 95% CI = [0.0003 – 0.0722], p = 0.048) and made fewer correct responses on the digit-symbol coding task (B =-0.8549, 95% CI =[-1.4616 – -0.2482], p =0.006). However, no effect of perceived risk was observed for our measure of proactive control (PBI; B=0.0003, 95% CI =[-0.0258 – 0.0264], p=0.982) nor overall risky choice (B=-0.1459, 95% CI =[-0.3497 – 0.0579], p =0.160). Thus, while it seems like overall perceived risk for contracting a severe case of COVID-19 predicts some variance in processing speed independently of pandemic-worry (i.e., FCQ scores), it seems this is not the case for either proactive control or overall risk-preferences. Considering these results, we have updated the manuscript to include this important distinction between worry and perceived risk.

We have updated the Methods, Results, and Discussion sections to explicitly state the question used and the observed relationships:

Page 17:

Finally, participants were asked to indicate if they felt sick with any COVID-19 symptoms and to determine if they believed themselves to be at an elevated risk of contracting a more severe case of COVID-19 (i.e., “Do you think you are at increased risk of experiencing a more severe case of COVID-19/coronavirus (i.e,. due to an underlying medical condition)?”) [53].

 Page 19:

Additionally, perceiving oneself as being at an elevated risk for contracting COVID-19 was associated with higher levels of pandemic-related worry (B =2.3799, 95% CI =[1.5289 – 3.2309], p <0.001) [53].

Page 20:

Interestingly, we observed that higher levels of pandemic-related worry—indexed by the FCQ—predicted lower performance (Fig 2B). Statistically, this predictive effect was significant, controlling for age, gender, overall stress, financial strain, perceived risk of contracting COVID-19, and several other demographic variables (FCQ = -0.940, CI = [-1.591, -0.289], p = .005; see S1 Table). Additionally, we found that even while controlling for pandemic-related worry, perceived risk of contracting a severe case of COVID-19 was also predictive of slower overall processing speed ( = 0.0362, 95% CI = [0.0003 – 0.0722], p = 0.048; see S1 Table). We also examined whether the predictive effect of worry on digit-symbol performance could be explained by a decrease in response caution, as indexed by faster correct RTs [42]. Instead, we found that increased FCQ scores were associated with slower correct response times (RTs; = 0.019, CI = [0.005, 0.033], p = .009; see Fig 2C and S2 Table) after controlling for the covariates listed above, indicating that the onus of worry’s predictive effect was a general slowing of information processing. 

 Page 22:

We next examined whether pandemic-related worry predicted deficits in the proactive behavioural index (PBI)[48]—an RT-based measure of proactive control, calculated as (RTAY - RTBX ) / (RTAY + RTBX )—finding that individuals with higher FCQ scores had lower PBIs (Fig 4B; = -0.0020, CI = [-0.0034, -0.0005] , p = 0.010; see S9 Table), controlling for gender, age, perceived risk of contracting COVID-19, self-reported, and financial stress (see S8 Table for full list of covariates).

 Page 23:

Importantly, this relationship between sensitivity to outcome probabilities and individual worry was robust after controlling for perceived risk of contracting COVID-19 (see S12 Table for full list of covariates).

 Page 30:

For instance, though we did not measure current compliance with COVID-19-related measures, it is possible that the behavioral repertoires examined here—executive functioning and risky decision-making—and the affective states (i.e., worry) of an individual might play a role in that individuals’ understanding and compliance with public health measures. Indeed, recent work has observed a positive relationship between one’s decision to comply with government sanctioned social-distancing directives and working-memory [76] as well as perceived (but not actual) pandemic-related risk [53]. Mirroring these findings, we found individuals who perceived themselves at higher risk for contracting COVID-19 also experienced greater pandemic-worry, and slower information processing (see S1 Table).

Given that the paper framed COVID-related changes in cognitive performance in terms of understanding and complying with public health measures, I thought the authors could have discussed current public failures to comply with such measures in a bit more depth and nuance. Are the authors thinking of cognitive performance as a primary driver of such compliance, or other factors as well? Perhaps the observed decline in COVID worry that they observed over time continued past June 2020, and can be thought of as an affective factor contributing to failures in public compliance to COVID safety measures. I thought the paper could do a better job discussing potential contributions of such factors to public behavior as well as making clear the limitations of the present data and what kinds of contextual factors future studies might want to consider when examining cognitive performance under duress.

We agree with the reviewer that the discussion of the possible future directions on the factors contributing to adaptive behaviour (e.g., following public health directive) under duress was unclear. As the reviewer points out, it is likely that both cognitive ability, and affective states contribute to one’s ability to respond to stressful situations (e.g., a global pandemic). However, the directionality of these effects remains unclear. We have accordingly updated the Discussion to reflect our speculations on the underlying factors contributing to adaptive behaviour and to further outline limitations of the present study.

Page 28: 

We believe the results outlined here have broader implications for human behaviour, as intact executive control is a cornerstone of healthy daily living (e.g., economic productivity), workplace performance [74], and the delay of gratification [75]. For instance, though we did not measure current compliance with COVID-19-related measures, it is possible that the behavioral repertoires examined here—executive functioning and risky decision-making—and the affective states (i.e., worry) of an individual might play a role in that individuals’ understanding and compliance with public health measures. Indeed, recent work has observed a positive relationship between one’s decision to comply with government sanctioned social-distancing directives and working-memory [76] as well as perceived (but not actual) pandemic-related risk [53]. Mirroring these findings, we found individuals who perceived themselves at higher risk for contracting COVID-19 also experienced greater pandemic-worry, and slower information processing (see S1 Table). Given the putative role of coronavirus anxiety in successful goal-directed behaviour, it is possible that an individual’s worry level plays a key role in the tendency to comply with public health measures: anxiety is associated both with worse working-memory and higher perceived risk for contracting COVID-19. One possibility is that pandemic-related worry could decrease compliance via putative changes in cognitive performance; another possibility is that worry could lead to increased compliance merely as a response to higher perceived risk of becoming sick, i.e. independently of the effects of worry on cognition. Given our cross-sectional design our results cannot shed light on the possible directionality of these effects. Future research should examine the underlying factors contributing to cognitive function and decision-making under high stress to better understand the relationship between affective and cognitive states, and the impact of these states on real-world behaviours. This is of particular interest considering the declining adherence to government directives and the far-from-universal acceptance of vaccination [6,7].

Reviewer #2: 

This research examined the relationship between Covid-19 related worry, working memory, and risk-taking behavior in an online sample. In general, I found the paper quite enjoyable — the abstract is exceptionally well-written. I think this paper would merit publication with minor revisions.

We are excited to hear the reviewer’s positive evaluation of our paper and are appreciative for their helpful and insightful comments.

1. The primary thing I would like to see in a revised manuscript would be the correlation of the cognitive tasks with one another. This seems important particularly in light of the claims in the discussion that they perhaps differentially rely on working memory.

We share the reviewers’ curiosity about whether our cognitive measures index shared versus disparate aspects of executive functions. While previous works has more extensively looked at the interrelationships between different cognitive tasks (see. Miyake et al. 2000, in Cognitive psychology), we report here correlations between the different measures in question. As suggested, we ran Pearson correlations between the different metrics of task performance: processing speed was computed as the total number of correct responses within 90 seconds on the digit symbol task, proactive control was computed from performance on the PBI as the relative degree of interference on trials recruiting proactive vs reactive control (RTAY – RTBX)/(RTAY + RTBX), and task switching ability was computed from the empirical bayes estimates of switch costs estimated using the linear hierarchical regression predicting log RTs. As seen below, we observe a statistically significant relationship between our measure of processing speed and both proactive control, and task-switching ability. However, reliance on proactive control was not related to performance on the task-switching paradigm. In keeping with the different relationships between the tasks and worry, this pattern of correlations further supports the possibility that these different measures of cognitive function index related, but distinct facets of executive functioning (Miyake et al. 2000). While processing speed seems to be a more fundamental aspect of cognition—being related to performance of the other two tasks—proactive cognitive control and task-switching performance seem to be unrelated. Perhaps this lack of correlation, despite the large sample size, reflects a differential reliance on working-memory capacity between the tasks. While these correlations do not provide evidence for a specific underlying factor structure, we believe they support the hypothesized effects of worry on working memory capacity outlined in the discussion. We have thus updated the discussion to reflect these analyses. 

 Switch Costs Digit Symbol Score PBI

Switch Costs 

Digit Symbol Score 0.112* 

PBI 0.002 0.124* 

Computed correlation used pearson-method with listwise-deletion.

* p < .05

Page 27:

Critically, working-memory demands are thought to play a key role in anxiety-induced impairments in set shifting ability, as both task complexity [64], and response preparation [65] have been found to mediate the predictive effects of anxiety on task-switching. Thus, it is possible that the chosen task switching paradigm did not sufficiently tax working memory to observe a worry-induced decline in performance, as task complexity and response preparation were not manipulated. Dovetailing with these findings, here we found that processing speed was related to performance on both the task-switching and proactive control tasks, but proactive control and task-switching performance did not appear to relate to one another (see S17 Table).

2. There is some use of acronyms without specifying what they are to the reader (FCQ; AX-CPT). I inferred, but it would be helpful to make it more explicit.

We thank the reviewer for their astute observation. We have updated the manuscript to include explicit definitions of the acronyms used.

3. I’m confused about the analysis outlined on p. 10. I thought that covid related worry was predicting cognitive function, but it sounds like those were used as predictor variables, given that they are described as fixed and random effects?

The reviewer is correct in pointing out that pandemic-related worry served as a predictor variable in the hierarchical models with task performance (i.e., correct responses, or response-times) as the outcome variable. The variables which served as both fixed and random effects were the experimental trial-by-trial manipulation. In other words, the outcomes were not the trial type (i.e., switch or repeat) but response-times, which were modeled as a function of trial type. We apologize for the confusion and have updated the manuscript to clarify these ambiguities in our analysis.

Page 17:

For the multi-level models, we included the within-participant trial-by-trial predictor variables as both random and fixed effects (i.e. task switch versus repetition, response congruency, decision frame losses versus gains, outcome probability likely versus unlikely). This allowed us to estimate group- and participant- level estimates for the effects in question (e.g., switch costs). 

4. Why was the Digit-Symbol Coding task run twice? I understand its due to accuracy, but aren’t they worried about practice effects? How many participants fell into this category? Were there similar accuracy constraints on the other tasks (it seems like no, why not?)

We share the reviewer’s concern over the decision to run the digit-symbol task twice. This procedure was chosen to mirror the procedure used in a previous study in our lab (da Silva Castanheira & Otto, submitted). Originally, this choice was used as an attention check to ensure participants were taking the task seriously, as the task was administered online. So as not to introduce new sources of variance, we decided to keep this procedure when collecting data during the three waves. 

The digit symbol coding task itself is relatively simple and does not require memorizing or learning any rules; the digit-symbol mappings are presented to the participant and do not change throughout the task (Mathias et al. 2017). This is because the digit-symbol coding task is specifically aimed at indexing individuals’ ability to process information quickly, and not their ability to learn, or their working memory capacity (Salthouse 1985). Thus, given that participants are given ample opportunity to practice, it is unlikely their performance reflects learning. 

Approximately 35% of participants were asked to repeat the task (28% in wave 1, 33% in wave 2, and 42% in wave 3). Since we anticipated worse performance on the tasks due to pandemic-induced worry and overall stress, we wanted to include as many opportunities as possible to ensure optimal task performance. Despite this, we found participants are overall worse at performing the task, both as a function of pandemic worry, and across waves (both pre-pandemic and during). Overall, we found that participants who repeated the task were generally worse—making fewer correct responses within the allotted 90 minutes—than those who did not ( = -2.3263, CI = [-4.0692, -0.5833], p =0.009), suggesting that task repetitions did not improve performance through practice effects. Critically, controlling for whether the participant repeated or did not repeat the task did not change the conclusion of our analysis: pandemic-related anxiety is associated with slowed processing speed ( = -0.8950, CI= [-1.5451, -0.2448[, p= 0.007). Thus, we are confident that this design choice did not fundamentally influence our results.

Accordingly, we have updated the description and logic behind this choice in our revised Methods.

 Page 8:

In keeping with previous pre-pandemic samples collected in the lab, we implemented an attention check designed for online data collection where participants who did not achieve 70% accuracy on the task were asked to complete the task a second time (28% of participants in wave 1, 33% in wave 2, and 42% in wave 3). In this case, only data from the second run were analyzed [44]. We compared the current samples to the previously collected data [44].

5. The claim that worry serves as a mediator to working memory disruption and correspondingly, obeying government sanctions is a big one. I would back off of that language, especially given that the link between worry, working memory, and Covid-19 related behavior were not tested in the scope of this paper.

We agree with the reviewer that the discussion of the results needs to be more carefully framed. Given Reviewer 1’s interest in this possible mediation (see comment 4), we have tempered language used in our discussion of the possible linkage between these constructs. In response to this comment, we have also elaborated on the limitations of this sort of interpretation and highlight the need for future research to directly test this question.

Page 28.

We believe the results outlined here have broader implications for human behaviour, as intact executive control is a cornerstone of healthy daily living (e.g., economic productivity), workplace performance [74], and the delay of gratification [75]. For instance, though we did not measure current compliance with COVID-19-related measures, it is possible that the behavioral repertoires examined here—executive functioning and risky decision-making—and the affective states (i.e., worry) of an individual might play a role in that individuals’ understanding and compliance with public health measures. Indeed, recent work has observed a positive relationship between one’s decision to comply with government sanctioned social-distancing directives and working-memory [76] as well as perceived (but not actual) pandemic-related risk [53]. Mirroring these findings, we found individuals who perceived themselves at higher risk for contracting COVID-19 also experienced greater pandemic-worry, and slower information processing (see S1 Table). Given the putative role of coronavirus anxiety in successful goal-directed behaviour, it is possible that an individual’s worry level plays a key role in the tendency to comply with public health measures: anxiety is associated both with worse working-memory and higher perceived risk for contracting COVID-19. One possibility is that pandemic-related worry could decrease compliance via putative changes in cognitive performance; another possibility is that worry could lead to increased compliance merely as a response to higher perceived risk of becoming sick, i.e. independently of the effects of worry on cognition. Given our cross-sectional design our results cannot shed light on the possible directionality of these effects. Future research should examine the underlying factors contributing to cognitive function and decision-making under high stress to better understand the relationship between affective and cognitive states, and the impact of these states on real-world behaviours. This is of particular interest considering the declining adherence to government directives and the far-from-universal acceptance of vaccination [6,7].

6. There are a number of typos/omissions in the paper, and I would suggest a careful read before the next revision. I caught a few, listed below:

- “Dependant” is a typo on p. 10

- Figure 1, the caption is missing another variable name when it says “overall perceived stress did not vary as a function of”

- Figure 2E is missing the bars

- Figure 5e, the x axis is “Pandemic-related worry”

- Pg 19 is missing an of in “the effects negative stress on task-switching ability”

We have updated the manuscript according to the comments above and thank the reviewer for their astuteness.

---

## [Decision Letter · Decision Letter 1]

2 Nov 2021

The impact of pandemic-related worry on cognitive functioning and risk-taking

PONE-D-21-23700R1

Dear Dr. da Silva Castanheira,

We’re pleased to inform you that your manuscript has been judged scientifically suitable for publication and will be formally accepted for publication once it meets all outstanding technical requirements. Although I was not able to secure a re-review from one of the original reviewers, I believe the comments were addressed and do not require an additional reviewer. 

Kind regards,

David V. Smith, Ph.D.

Academic Editor

PLOS ONE

Additional Editor Comments (optional):

Reviewers' comments:

Reviewer's Responses to Questions

**Comments to the Author**

1. If the authors have adequately addressed your comments raised in a previous round of review and you feel that this manuscript is now acceptable for publication, you may indicate that here to bypass the “Comments to the Author” section, enter your conflict of interest statement in the “Confidential to Editor” section, and submit your "Accept" recommendation.

Reviewer #1: All comments have been addressed

2. Is the manuscript technically sound, and do the data support the conclusions?

Reviewer #1: Yes

3. Has the statistical analysis been performed appropriately and rigorously? 

Reviewer #1: Yes

4. Have the authors made all data underlying the findings in their manuscript fully available?

Reviewer #1: Yes

5. Is the manuscript presented in an intelligible fashion and written in standard English?

Reviewer #1: Yes

6. Review Comments to the Author

Reviewer #1: Thank you. The authors have addressed my concerns.

Very minor revision - I note that in the revised text the authors use the acronym AMT (page 29) which does not appear to be used anywhere else in the text. With some sleuthing I was able to determine this means Amazon Mechanical Turk, but I think this term could be corrected to “MTurk” as that shortened name is used elsewhere in the manuscript, or the acronym should be defined appropriately.

7. PLOS authors have the option to publish the peer review history of their article (what does this mean?). If published, this will include your full peer review and any attached files.

Reviewer #1: No

---

## [Editor Report · Acceptance letter]

8 Nov 2021

PONE-D-21-23700R1 

The impact of pandemic-related worry on cognitive functioning and risk-taking 

Dear Dr. da Silva Castanheira:

I'm pleased to inform you that your manuscript has been deemed suitable for publication in PLOS ONE. Congratulations! Your manuscript is now with our production department. 

Kind regards, 

on behalf of

Dr. David V. Smith 

Academic Editor

PLOS ONE